# The Multiple Faces of the MRN Complex: Roles in Medulloblastoma and Beyond

**DOI:** 10.3390/cancers15143599

**Published:** 2023-07-13

**Authors:** Marialaura Petroni, Veronica La Monica, Francesca Fabretti, Mariaconcetta Augusto, Damiana Battaglini, Francesca Polonara, Stefano Di Giulio, Giuseppe Giannini

**Affiliations:** 1Department of Molecular Medicine, University La Sapienza, 00161 Rome, Italy; veronica.lamonica@uniroma1.it (V.L.M.); francesca.fabretti@uniroma1.it (F.F.); mariaconcetta.augusto@uniroma1.it (M.A.); damiana.battaglini@uniroma1.it (D.B.); polonara.1955470@studenti.uniroma1.it (F.P.); stefano.digiulio@uniroma1.it (S.D.G.); giuseppe.giannini@uniroma1.it (G.G.); 2Istituto Pasteur-Fondazione Cenci Bolognetti, 00161 Rome, Italy; 3Center for Life Nano- & Neuro-Science, Istituto Italiano di Tecnologia (IIT), 00161 Rome, Italy

**Keywords:** MRN complex, DDR, RSR, haploinsufficiency, p53, synthetic lethality, target therapy

## Abstract

**Simple Summary:**

Defects in *MRE11A/RAD50/NBS1* (*MRN*) genes and overexpression of the MRN proteins are both linked to cancer, generating confusion about how the MRN complex impacts cancer initiation and progression. In this review, we examined the most relevant studies on the topic, taking advantage of our recent publication on the role of *NBS1* as both a haploinsufficient tumor suppressor and an essential gene in SHH-medulloblastoma. Finally, we discussed the possibility to use genomic and molecular characterization of the MRN complex in clinical practice to improve therapeutic strategies, together with future research directions and challenges.

**Abstract:**

Hypomorphic mutations in MRN complex genes are frequently found in cancer, supporting their role as oncosuppressors. However, unlike canonical oncosuppressors, MRN proteins are often overexpressed in tumor tissues, where they actively work to counteract DSBs induced by both oncogene-dependent RS and radio-chemotherapy. Moreover, at the same time, *MRN* genes are also essential genes, since the constitutive KO of each component leads to embryonic lethality. Therefore, even though it is paradoxical, *MRN* genes may work as oncosuppressive, oncopromoting, and essential genes. In this review, we discussed how alterations in the MRN complex impact the physiopathology of cancer, in light of our recent discoveries on the gene–dosage-dependent effect of NBS1 in Medulloblastoma. These updates aim to understand whether MRN complex can be realistically used as a prognostic/predictive marker and/or as a therapeutic target for the treatment of cancer patients in the future.

## 1. Introduction

DNA double-strand breaks (DSBs) are highly toxic DNA lesions that can lead to loss of genes, chromosomal instability, and cancer. DSBs can be induced by damaging agents but also arise during physiological cellular processes, such as replication, recombination in meiosis, and rearrangements of immunoglobulin (Ig) genes. To repair DSBs and restore genomic integrity, eukaryotic cells have developed several pathways of DNA damage response (DDR) in which the MRE11/RAD50/NBS1 (MRN) complex is a shared and essential player. It consists of highly conserved proteins of which MRE11 and RAD50 can be found even in lower eukaryotes [1]. The importance of the MRN complex is highlighted by the fact that nullizygosity causes early embryonic lethality in mice [2,3,4].

So far, mutations in all three components of the MRN complex have been reported in humans leading to different diseases. In particular, hypomorphic mutations in *MRE11A* lead to ataxia telangiectasia-like disease (ATLD-OMIM: 604391), a rare disorder characterized by progressive cerebellar ataxia, dysarthria, abnormal eye movements, and absence of telangiectasia. ATLD patients show normal levels of total IgG, IgA, and IgM, although there may be reduced levels of specific functional antibodies. At the cellular level, ATLD patients exhibit hypersensitivity to ionizing radiation and radioresistant DNA synthesis. Destabilization of all three members of the MRN complex has been reported in ATLD type 1 patients [5,6,7]. Importantly, biallelic *MRE11A* mutations have also been reported in two unrelated individuals sharing severe primary microcephaly, moderate to severe intellectual disability, and pyramidal signs, but no cerebellar atrophy, nor immunodeficiency [8]. Both patients were compound heterozygotes for a truncating or missense mutation and carried a translationally silent mutation resulting in reduced but normal MRE11 protein [8].

Hypomorphic mutations in *NBS1* (officially known as *NBN*) cause the Nijmegen breakage syndrome (NBS-OMIM: 251260), an autosomal-recessive disorder characterized by progressive microcephaly, mild to moderate growth retardation, dysmorphic facial features, immunodeficiency, and predisposition to cancer, particularly to lymphoid malignancies. Germline defects in NBS patients include infertility and compromised sexual maturation. At the cellular level, NBS is characterized by radiosensitivity, chromosomal breakage, and defective cell cycle checkpoints. Ninety percent of cases are caused by the Slavic founder mutation, 657del5 (ACAAA) (rs587776650), within the *NBS1* gene. This variant leads to the expression of two fragments of 26 and 70 kDa (p26 and p70, respectively), the latter resulting from alternative translation initiation, maintaining binding ability to MRE11 [9]. These fragments retain residual function but are much less expressed [10].

Hypomorphic mutations in *RAD50* lead to NBS-like disorder (NBSLD-OMIM: 613078) characterized by facial features resembling NBS, severe prenatal growth retardation and persistent postnatal growth restriction, congenital microcephaly, mild to borderline intellectual disability, normal sexual development, radioresistant DNA synthesis, and no immunodeficiency or myelodysplasia or early neurodegeneration as key features. So far, only two patients have been reported, one with biallelic mutations leading to a RAD50 protein deficiency [11] and one with a novel homozygous variant, c.2524G > A, leading to premature protein truncation and thereby, most likely, to a loss of RAD50 function [12].

The broad clinical spectrum of patients with MRN–related disorders demonstrates how the MRN complex plays an essential role in cellular homeostasis. This review aims to recapitulate what we know about the multiple functions of the MRN complex, their impact on cancer progression and resistance, and how far we are from the possibility to use these notions to improve prognosis and therapy in cancer patients.

## 2. MRN Complex Functions in Cellular Homeostasis

In the following paragraph, we will discuss the multiple functions of the MRN complex. As expected, most of them are attributable to the pivotal role of the MRN complex in DDR (Figure 1).

### 2.1. MRN Complex in DSBs and Replication Stress Handling

The MRN complex functions as a sensor of DSBs but also directly in the repair of DNA damage through two major pathways of DNA damage response: homologous recombination repair (HRR) and non-homologous end joining (NHEJ), the latter, error-prone [13]. How exactly the MRN complex participates in the DNA damage response (DDR) has been extensively studied and reviewed in [14]. Briefly, following the DNA damage, the MRN complex is rapidly recruited to DNA damage sites by γ-H2AX and RAD17 [15]. Once bound to DSBs, the MRN complex can recruit and activate various DDR proteins, including ATM [16], and Bloom syndrome protein (BLM) [17]. Next, ATM in turn phosphorylates several targets including MRN complex proteins to start downstream signaling pathways. The MRN complex is required for the activation of ATM as supported by biochemical and genetic analyses in mice, yeast, and human cells [18,19,20,21]. MRN also orchestrates the pathway choice in DSB repair. Indeed, the MRN complex is essential in the formation of single-stranded DNA overhang that serves as a platform for the recruitment of HR repair proteins. First, MRE11 endonuclease and C-terminal binding protein (CtBP)-interacting protein (CtIP) generate an initial resection nick. After that, the exonuclease activities of MRE11 and EXO1/BLM (exonuclease 1/BLM RecQ-like helicase) bidirectionally resect toward and away from the DNA end, which commits to HR and disfavors NHEJ [22]. A recent paper demonstrated that the MRN complex also influences the DSB repair pathway choice by interacting with DNA-dependent protein kinase (DNA-PK). In cases where NHEJ fails due to incompatible DNA ends, DNA-PK promotes DNA end resection by stimulating MRE11 endonuclease activity to remove the complex, triggering further DNA end resection and repair by HR [23].

A major source of endogenous DNA damage in our cells is represented by replication stress (RS) [24,25] that, if not correctly handled, may cause genome instability. Cellular response to RS activates checkpoints, primarily mediated by the kinase ATR, to arrest the cell cycle and repair DNA damage. Accumulating evidence demonstrates that the MRN complex entitles several roles in the replication stress response (RSR). The MRN complex can bind DNA structures encountered at stalled or collapsed replication forks (such as ssDNA and dsDNA junctions or breaks); activate ATR; and promote the restart of DNA replication [26,27]. Moreover, MRN-mediated resection of nascent DNA likely frees the replisome from the collapsed RF and generates 3′ overhangs for initiation of HR, which in turn restores the replication under stress [28]. In vitro experiments have shown that MRE11 nuclease activity can process replication structures to form ssDNA gaps behind forks, particularly in the absence of protection from RAD51 [29] or PARP1 and BRCA1 [30], suggesting that the MRN complex may play a role at stalled replication forks also in HR and BER defective cells. Therefore, the MRN complex redistribution to restarting forks prevents the accumulation of chromosomal abnormalities during DNA replication [31].

Additionally, the MRN complex regulates fork progression, altering the protein landscape on nascent DNA by a cGAS-STING-TbK1 pathway-dependent modulation of Interferon stimulated gene 15 (ISG15) [32,33].

Importantly, topoisomerase activity generates DNA breaks as normal intermediates during chromosome metabolism. MRN complex prevents topoisomerase-mediated DSBs by removing TOP1 and TOP2 trapped on DNA [34,35,36,37].

Since DSB accumulation may favor tumor initiation but, at the same time, if strongly stimulated, represents, currently, the main therapeutic strategy to treat tumor cells, it is supposed that MRN complex expression and functionality may impact cancer development and progression as much as therapy effectiveness.

#### 2.1.1. MRN Complex Defects as Risk Factor for Cancer Development and Assumption for a Synthetic Lethal-Based Strategy

There is accumulating evidence that ATR/ATM-regulated DDR may serve as an inducible barrier to constrain tumor development in its early, pre-invasive stages, by inducing cell death or senescence [38]. As the MRN complex plays a crucial role in ATR and ATM activation in response to RS and DSBs, respectively [18,19,20,21,39], MRN complex inactivation may disturb the ATR/ATM-dependent anti-cancer barrier, leading to cancer progression. Coherently with this idea, patients suffering from ATLD, NBS, and NBSLD are typically characterized by a decrease in ATM activation or activity [12,16], genome instability, and cancer predisposition [40,41].

Importantly, epidemiological and clinical studies have found that heterozygous NBS patients, who are clinically asymptomatic, also display an elevated risk to develop some types of malignant tumors [42], supporting the idea that *NBS1* gene is a haploinsufficient oncosuppressor. Moreover, heterozygous NBS patients showed increased frequency of chromosomal translocations [43]. Thus, DDR haploinsufficiency is supposed to be the pathogenic mechanism through which *NBS1* heterozygosity predisposes cells to malignancy. Consistently, studies in *Nbn*^+/−^ mice showed a significantly increased occurrence of spontaneous solid tumors (affecting the liver, mammary gland, prostate, lung, and lymphocytes) without loss of heterozygosity (LOH), providing a clear relationship between *NBS1* heterozygosity and increased cancer risk [44]. Despite studies evaluating the frequency of inherited gene mutations in populations with cancer family history may be considered questionable, due to the potential involvement of other genes segregating with *NBS1* locus, other evidence supports the concept that NBS1 is a haploinsufficient oncosuppressor. Indeed, several epidemiological studies report an increased frequency of *NBS1* heterozygous germline mutations in patients with sporadic tumors compared to the control population (see Table 1 and Figure 2).

Interestingly, a significant correlation was also found between *RAD50* heterozygous germline mutations and sporadic cancer (see Table 2 and Figure 2). Unfortunately, studies on *MRE11A* gene alterations in cancer are few, and most of them do not have control groups limiting any reliable conclusion (see Table 3). However, considering the studies with the control population, it seems that *MRE11A* heterozygous germline mutations do not predispose to cancer, maybe suggesting that *MRE11A* mutations are less tolerated by the cells or do not offer any survival/proliferative advantage.

Importantly, NBS1 is a member of the MRN complex more frequently mutated in cancer. Currently, we cannot exclude that this may be associated with MRN complex-independent roles. Indeed, it is possible that, in NBS1-defective patients, immunodeficiency may lead pre-cancerous cells to escape from “immune surveillance” [45], contributing to a higher rate of tumor development.

Defects in one DNA repair pathway can be compensated for by other DNA repair pathways. Such compensating pathways can be identified in synthetic lethality screens and then specifically targeted for the treatment of DNA repair-defective tumors. Coherently with this idea, several studies support the possibility to use deficiencies in the MRN complex-dependent DDR as an opportunity for therapeutic exploitations in those tumors where the MRN complex is frequently dysfunctional. In vitro experiments demonstrate that deficiency in HR by mutations in the *MRN* genes may sensitize endometrial [46] and microsatellite unstable colorectal cancer cells [47,48] to treatment with poly (ADP-ribose) polymerase (PARP) inhibitors and might therefore serve as a predictive biomarker of PARP inhibitor therapy. Microsatellite instability-dependent mutations in *CtIP* and *MRE11A* confer hypersensitivity to PARP inhibitors in myeloid malignancies [49]. Takagi et al. found SNVs and/or copy number alterations in DDR-associated genes, including *MRE11A*, in 48.4% of all neuroblastoma analyzed, and these data could explain the high sensitivity to Olaparib observed by the authors in most NB cells, in vitro [50]. Both MRE11 and NBS1 depletion increases sensitivity to PARP-1 inhibitor KU 58948 in breast cancer cell lines [51]. Similarly, MRE11 dysfunction is associated with increased sensitivity to DNA-damaging therapy and inhibitors of ataxia telangiectasia and Rad3-related (ATR) and PARP but not to the anti-microtubule agent Taxol, supporting that truncated or missense variants of *MRE11A* may promote hypersensitivity to DNA-damaging therapeutics in breast cancer [52]. Coherently, another study demonstrates that *MRE11A* and *NBS1* transcript levels associate with resistance to Olaparib in breast cancer cell lines [53]. Thus, since MRN defects commonly occur in ER/PR/ERBB2 (triple)-negative breast carcinomas (TNBC), it is expected that PARP inhibitors could be useful in the treatment of this group of patients, presently the most difficult-to-treat subset of breast tumor patients. In Head and neck squamous cell carcinoma (HNSCC), the downregulation of the MRN complex combined with PARPi, leads to accumulation of lethal DNA DSBs in vitro and in vivo [54]. In epithelial ovarian cancer, where a lack of MRN complex protein detection was seen in 41% of the tumors, MRE11 knockdown increased sensitivity towards the PARP inhibitor BMN673 [55]. Moreover, MRE11 inhibition was synthetically lethal in platinum-sensitive XRCC1-deficient ovarian cancer cells and 3D-spheroids [56].

#### 2.1.2. MRN Complex as a Critical Factor in Resistance to Oncogene- and Therapy-Induced DSBs/RS

Overexpression or constitutive activation of oncogenes, such as H/KRAS, c-Myc, NMYC, MDM2, BCL-2, and cyclin E, causes alterations of replication timing and progression leading to RS [57]. Therefore, RS is linked to pre-tumor [58,59,60] and tumor [61,62,63,64,65,66,67,68]. Extensive literature supports the concept that cells proliferating under the pressure of oncogenes need to counteract the RS by strengthening the RSR. Accordingly, *MRN* genes are often overexpressed in tumor tissues [69,70,71,72,73], also because they are transcriptional targets of some oncogenes and proto-oncogenes. In particular, it has been demonstrated that c-Myc transcriptionally regulates *NBS1* [74] and we published that MYCN transcriptionally regulates all three of the *MRN* genes [75]. Coherently with the idea that MRN complex is necessary to counteract oncogene-induced RS, MRE11 inactivation leads to enhanced DNA damage, decreases survival, and increases apoptosis in cells with exogenous overexpression of c-Myc or MYCN [69,73]. Accordingly, we also published that genetic or pharmacological inhibition of MRE11 led to the accumulation of RS and DDR markers and caused p53-dependent cell death in MYCN-amplified neuroblastoma, in vitro and in pre-clinical models [73]. Moreover, hypomorphic *MRE11A* increases levels of oncogene-induced DNA damage, R-loop accumulation, and chromosomal instability in a breast tumorigenesis mouse model [52].

Primary anti-cancer therapies, such as ionizing radiation and chemotherapeutic agents, induce cell death by directly or indirectly causing RS and DNA damage. Inherent resistance of tumors to DNA damage often limits the therapeutic efficacy of these agents, becoming the leading cause of treatment failure in cancer patients. Possibly due to the roles of the MRN complex in DDR and RSR [75,76], most studies (with one only exception [70]) support the idea that the MRN complex is a critical factor in driving chemo- and radio-resistance. In particular, nuclear accumulation of the MRN complex was associated with chemo-resistance in gastric cancer [77] and ovarian cancer [56,78], and high expression of RAD50 correlates with radio-resistance in colorectal cancer (CRC) patients [79]. Moreover, in pre-clinic ovarian cancer studies, RAD50 expression resulted higher in platinum-resistant cells [80], and platinum treatment increased nuclear sub-cellular localization of NBS1 [78] and RAD50 [80] more in platinum-resistant cells with respect to the sensitive ones, supporting the idea that accumulation of nuclear MRN proteins could contribute to cisplatin resistance in this tumor background. Coherently, it has been found that MRN targeting may also rescue radioresistance. Indeed, overexpression enhances radio-resistance in non-small cell lung cancer (NSCLC), in vitro [81].

With a similar rationale, other studies showed that MRN complex targeting sensitizes to chemo- and radiotherapy. In particular, RAD50 depletion sensitizes to radiotherapy human nasopharyngeal carcinoma [82] and NSCLC [81] cells. Moreover, the expression of a mutant *NBS1* by a dominant negative recombinant adenoviral construct significantly increases cisplatin-induced DNA DSBs and cytotoxicity in HNSCC cell lines [83]. Cells deficient in MRE11 or expressing the nuclease-deficient form of MRE11 resulted in more sensitivity to Etoposide [37]. Moreover, a combination of mirin, an inhibitor for the MRN complex [84], with ionizing radiation treatment significantly enhanced DSBs reduced clonogenic cell survival, inhibited cell proliferation, and promoted cell apoptosis in esophageal squamous cell carcinoma (ESCC) cells [85].

Interestingly, several authors find that cancer stem cells (CSC) from different tissues efficiently resolve RS [86,87,88]. Regarding to this, it has been demonstrated that inhibition of MRE11 and RAD51 effectively kills colorectal CSC resistant to irinotecan and ATR/CHK1 inhibitors via a mitotic catastrophe process, after RSR weakening and defective mitoses [89].

### 2.2. MRN Complex in Innate and Adaptive Immune Response

The appearance of dsDNA in the cytoplasm, which is normally a DNA-free environment, triggers potent inflammatory pathways that culminate in the production of interleukin 1β (IL-1β) and type 1 interferon. Such innate immune responses alert the host to the presence of danger and are important for the defense against viruses and bacteria [90].

Upon viral infection, viral DNA in the cytoplasm is perceived as a DSB, triggering endogenous ATM-dependent DNA damage response. The MRN complex has emerged as a detector of many different viral genomes that activates a localized ATM response that specifically prevents viral DNA replication [91]. Importantly, to avoid this response, several viral proteins were found to mediate proteasome-dependent degradation and mislocalization of components of the cellular DNA damage machinery, including the MRN complex [92,93,94,95]. In contrast, MRN was recruited within viral replication compartments and exerted a positive effect on herpes simplex virus 1 (HSV-1) lytic replication [96,97], suggesting that MRN complex could play different roles in Adeno-associated virus (AAV) replication depending on the helper virus that is present.

Recent studies demonstrated that, working as a sensor of cytosolic DNA, the MRN complex has a role in innate immune activation. In particular, cytoplasmic delivery of dsDNA by transfection of DNA or infection with a virus resulted in the formation of distinct dsDNA-RAD50-CARD9 complexes that selectively induced NF-κB signaling for IL-1β production [98]. It was also found that the physical interaction between MRE11 (or RAD50) with dsDNA in the cytoplasm was required for the stimulation of cGAS/STING-inflammatory signaling (IFN genes) [99]. Importantly, NBS1 is not essential for dsDNA-induced type I IFN production [99], but, favoring accumulation of the MRN complex in the nucleus, even decreases cytosolic DNA sensing by MRE11. More recently it has been demonstrated that also NBS1 binds to micronuclear DNA, probably via MDC1, and recruits ATM and CtIP to convert micronuclear dsDNA ends to ssDNA ends. This conversion prevents cGAS from binding to micronuclear DNA and avoids the cGAS-STING dependent induction of immune signaling and cellular senescence [100]. Therefore, NBS1 negatively regulates the MRE11-RAD50-dependent production of IFN genes both by the transport of MRE11 in the nucleus and by reducing dsDNA in the cytoplasm.

Programmed DSBs and repair systems such as V(D)J and class-switch recombination (CSR) are part of a developmental program that ensures the diversity of antigen receptors in B and T lymphocytes and the production of specific classes of immunoglobulin. Inability to persecute the programmed DDR results in the primary immunodeficiency (PID) [101]. Much evidence supports the role of NBS1 in both V(D)J [102] and CS recombination [103]. NBS1 localizes at switch regions in activated B cells [104] and *Nbn* conditional KO mice show impaired class-switch recombination [105]. Coherently, NBS patients have reduced titers of switched serum isotypes and altered switch junctions [106]. Interestingly, immunodeficiency is a hallmark of NBS but not of NBSLD and AT-LD patients. This aspect may suggest an MRN complex independent role of NBS1 in lymphoid cells.

### 2.3. MRN Complex in Telomere Homeostasis

Many DDR proteins, particularly those involved in responding to DNA DSBs, physically associate with telomeres and play key roles in their maintenance. In fact, components of the DDR are necessary both for normal telomere homeostasis and for responding to dysfunctional telomeres (Figure 3) [107]. The MRN complex associates with human telomeres through interaction with TRF2 and it is required for the activation of ATM at dysfunctional telomeres [108,109]. Moreover, the MRN complex participates in the formation of the structure of t-loops, the lariats formed through the strand invasion of the telomere terminus into the duplex telomeric DNA, which contributes to telomere protection, and consequently to the preservation of genome stability, by effectively shielding the chromosome ends from DNA damage response factors that interact with DNA ends [107,109]. Telomeres of telomerase-negative primary cells recruit MRE11, phosphorylated NBS1, and ATM in every G2 phase of the cell cycle, when telomeres are unprotected and accessible to modifying enzymes. Degradation of the MRN complex, as well as inhibition of ATM, led to telomere dysfunction. Therefore, a localized MRN complex-dependent DDR at telomeres after replication is essential for recruiting the processing machinery that promotes the formation of a chromosome end protection complex [110], avoiding chromosome instability and senescence.

### 2.4. MRN Complex in Centrosome Maintenance and Mitotic Spindle Dynamic

The accurate chromosomal segregation that occurs during cell division is controlled by the mitotic spindle and alterations in centrosome stability/number or mistakes in spindle formation/disassembly can result in aneuploidy or cytokinesis failure, which are linked to cancer [111,112]. Importantly, the MRN complex seems to be associated with both centrosome and mitotic spindle homeostasis. In particular, NBS1 localizes at the centrosome in mitosis and in interphase, and because its depletion induces centrosome hyper-duplication, it is widely accepted that its expression is essential for correct centrosome duplication [113]. Interestingly, MRE11 localizes both at centrosome and microtubules in mitotic cells [114], and its depletion triggers centrosome amplification [115]. Moreover, it was demonstrated that the MRN complex is necessary for RCC1-mediated RanGTP gradient, essential for the self-assembly of microtubules into a bipolar structure. Indeed, recent studies found that the inhibition of MRN complex function in mammalian cells, reducing RCC1 association with mitotic chromosomes, disrupting the RCC1-dependent RanGTP gradient, and triggering metaphase delay [116]. Moreover, Xu et al. found that the MRN complex together with MMAP (which forms a mitosis-specific complex named mMRN) regulates the PLK1–KIF2A signaling cascade, widely known for being involved in the microtubule depolarization and consequently in spindly disassembly [114].

Curiously, centrioles and centrosomes contain RNA but do not contain any DNA [117]; therefore, it is possible that the MRN complex regulates centrosome/spindle stability in a DDR-independent manner.

### 2.5. MRN Complex and Cancer-Related Pathways

Numerous studies indicate both inverse and positive correlations between MRN complex and mitogenic pathways expression. In particular, several authors found that NBS1 expression negatively regulates the NOTCH pathway. It was demonstrated that, independently of the DDR, NBS1 deletion up-regulates the NICD protein level, as well as NOTCH activity in neurons, causing in turn repression of neurite outgrowth and neuronal migration in post-mitotic neurons [118]. Moreover, in line with Cheung et al.’s study which found NOTCH pathway upregulation in lymphoblastoid cell lines from NBS heterozygous carriers [119], we recently published that *Nbn* heterozygosity increases the Notch pathway and a Notch-dependent clonogenicity in cerebellar granule cell progenitor (cGCPs) primary cultures, isolated from a Medulloblastoma (MB)-prone mouse model [120].

On the contrary, NBS1 expression positively regulates RAS/RAF/MEK/ERK cascade upon activation by the growth hormone IGF-1 [121]. Moreover, we recently found that NBS1 KO correlates with low levels of several SHH targets (GLI1 and MYCN) and with an impairment of SHH-dependent proliferation in cGCPs, in vivo and in vitro, and in murine MB primary cells [97], suggesting a link between NBS1 expression and the SHH pathway.

Despite a single exception in ESCC, where NBS1 expression inversely correlates with SNAIL expression and reduces E-cadherin expression [122], increasing literature supports a positive correlation between MRN complex expression and invasion/metastasis-related genes. Cristina Espinosa-Diez et al. demonstrated that in the presence of significant DNA damage, MRN complex targeting by miR-494 and miR99b decreases VEGF signaling and thereby angiogenesis, thus supporting the angiogenic role of the MRN complex in endothelial cells [123]. In papillary thyroid carcinoma (PTC), the downregulation of MRE11 and RAD50 expression, through the lncRNA SLC26A4-AS1-mediated disruption of the DDX5-E2F1 transcription factor complex, inhibits the invasion and metastasis capability of cancer cells [124]. In oral cancer, it was found that MRE11 promotes proliferation, epithelial–mesenchymal transition, and metastasis regulating RUNX2, CXCR4, AKT, and FOXA2 in a nuclease-independent manner [125]. Moreover, in breast cancer, MRE11 overexpression promotes proliferation by activation of the STAT3 signaling and its downstream effectors on one side, and tumor cell invasion and migration by an increase in secretion of metastasis-associated MMP proteins (MMP-2 and MMP-9), on the other side [126]. Overexpression of NBS1 induces EMT through the upregulation of the PI3-KINASE/AKT/SNAIL/MMP2 axis in HNSCC [127] and by MMP-2-independent expression of two heat shock proteins HSPA4 and HSPA14 in NSCLC [128]. In high-grade serous ovarian cancer, RAD50 overexpression activates NF-kB which increases several mesenchymal phenotype markers such as N-cadherin, Vimentin, SNAIL, and TWIST, and reduces the expression of epithelial phenotype marker E-cadherin [129]. Analysis of differentially expressed genes associated with low expression of *MRE11A*, *RAD50*, and *NBS1* identified an increased expression of genes associated with mitochondrial dysfunction and metabolic reprogramming that could contribute to the aggressive behavior of MRN-deficient tumors [130].

Importantly, cancer-related pathways are the most enriched pathways that were found in NBS-fibroblasts compared to healthy fibroblasts, as well as in NBS-iPSCs compared to embryonic stem cells [131].

## 3. The Prognostic Significance of the MRN Complex Expression in Cancer

Coherently with the oncogenic role of the MRN complex in protecting cancer cells from RS/DSBs accumulation, countless retrospective studies indicate that MRN complex overexpression is an independent marker of poor prognosis in cancer and predicts the effect of specific therapies (Figure 4).

In particular, high levels of NBS1 correlate with poor prognosis in advanced HNSCC [71] and with aggressive behavior (e.g., recurrence⁄/metastasis) in oral squamous cell carcinoma (OSCC) and NO-HNSCC [72]. In gastric cancer, nuclear accumulation of the MRN proteins was associated with cancer progression, reduced DNA damage, poor prognosis, and chemo-resistance [77]. In ovarian cancer, MRN gene or protein overexpression was more frequently found in high-grade tumors [55,132], significantly associated with platinum resistance, shorter progression-free survival (PFS), and poor overall survival (OS) [56,78,80], and was an independent prognostic factor for recurrence [133]. In uveal melanoma NBS1 expression strongly correlated with tumor severity and metastatic death and it was proposed to use a high expression of NBS1 at the time of diagnosis of the primary eye tumor to select those patients who are at high risk of metastasis, in order to treat them prophylactically with adjuvant systemic therapy [134]. Combined high expression of MRE11 with RAD51 or ATM results correlated with worse prognosis in metastatic/recurrent CRC patients undergoing oxaliplatin-based chemotherapy [135] and in rectal cancer patients, including a neoadjuvant radiotherapy sub-cohort [136], respectively. MRN expression is associated with higher histological tumor stage, worse DFS, and OS in DNA (MMR)-positive rectal cancer patients including the neoadjuvant radiotherapy subgroup [137]. In TNBC and BRCA group patients having low/medium expression levels of RAD50 were observed to survive longer than patients exhibiting high expression of RAD50 [138].

Unexpectedly, other studies report that overexpression of MRN genes/proteins is a marker of good prognosis (Figure 4). In particular, MRN complex expression seems to correlate with good prognosis in ESCC and it is linked to an increased response to neoadjuvant chemotherapy in biopsies taken from patients who underwent treatment with 5-fluorouracil and cisplatin [70]. In muscle-invasive bladder cancer high MRE11 expression is predictive of good response to radiotherapy but not to cystectomy and it is suggested to use MRE11 expression to select patients for radiotherapy or cystectomy [139,140,141]. In left-sided colon and rectal cancer (LSCRC) [142] and CRC [143], strong expression of MRN complex proteins was found related to microsatellite stability (MSS) and better OS. Moreover, low RAD50 expression was associated with worse disease-free survival and OS in early-stage (T1-2) and low-grade (G12) tumors in postoperative rectal samples [137] and poor prognosis in colorectal mucinous (MC) patients [144]. Again, in early breast cancer, weak expression of the MRN complex correlates with high histologic grade, estrogen receptor negativity, and poor survival [130,145].

Therefore, the prognostic significance of the MRN complex expression is currently controversial. Differences in stages of disease, measured endpoint (e.g., local recurrence vs. overall survival), tissue of tumor origin, and standardized criteria to evaluate MRN protein levels in tumor tissues [146] may contribute to generating conflicting results. Moreover, the prognostic role of the MRN complex may be tissue-specific and linked to certain combinations of chemotherapy, radiotherapy, and surgery, which vary among different cancers.

However, it is also possible that higher MRN levels correlate with good prognosis if compared with conditions in which lower MRN levels are due to gene defects which, perturbing complex functionality, contribute to genomic instability. A tumor model that may recapitulate this phenomenon is colon cancer. Here, *MRE11A* defects correlate with defects in mismatch repair that are expected to lead to microsatellite instability (MSI), a predictive marker for lack of efficacy of fluorouracil-based therapy [135]. This raises the possibility that some MRE11-negative patients have poor prognosis because of their MSI status. Another similar example is in neuroblastoma. Here, 11q deletion and *MYCN* amplification are very frequent (35–45% vs. 20–25%, respectively), mutually exclusive, and correlate with poor survival [147,148]. We published that *MRE11A* mRNA expression was significantly higher in worst prognosis cases characterized by *MYCN*-amplification compared to *MYCN* single copy (MNSC) [73]. However, when we focused our analysis on the MNSC subgroup, we found that a very low *MRE11A* expression was associated with poor survival and high-risk and stage [73]. Importantly, the 11q region contains the *MRE11A* locus. Therefore, the correlation between low expression of MRE11 and the worst prognosis in MNSC patients may be due to the loss and not to the downregulation of MRE11.

Moreover, truncating or missense variants of *MRN* genes result in proteins that, despite being overexpressed, may have lost their capability to restrain oncogene/therapy-induced RS and DSBs. Coherently, Na et al. reported that a C-terminally truncated MRE11 was responsible for the decreased HR repair and enhanced radiosensitivity in bladder cancer cells, suggesting that the increase in this truncated form of MRE11 might explain some of the paradoxical findings of high MRE11 expression associated with better survival rate after radiotherapy [149].

## 4. Medulloblastoma as a Model for the Pleiotropic Role of MRN Complex in Cancer

Despite paradoxical, collectively, clinical, preclinical, and in vitro studies indicating that, due to their multiple functions in DDR and beyond, both reduced and increased levels of MRN proteins may play a role in carcinogenesis and tumor progression. In order to validate this concept, we recently addressed the pleiotropic role of the MRN complex in cancer by using the model of Medulloblastoma (MB) [120], the most common malignant brain tumor affecting children [150]. Here, exploring the effects of different levels of NBS1, we discovered that alterations in the expression of the *MRN* genes, both in the sense of decrease and increase, impact cancer development and tumor full-blown survival/progression [120]. In particular, we found that mono-allelic KO of *Nbn*, reducing the total amount of NBS1 available for the cell, significantly increases the probability of developing a tumor in an SHH-MB-prone mouse model. These data, together with the fact that MBs were observed in NBS patients [151], and heterozygous germline or somatic mutations in *NBS1*, *RAD50*, and *MRE11A* were discovered in MB patients [152,153,154,155], suggest that *MRN* genes may work as haploinsufficient oncosuppressors in MB development. Coherently with the idea that *MRN* genes haploinsufficiency could favor genomic instability and consequently tumor proneness as an effect of defective DDR, we also demonstrated, for the first time, that *Nbn* is haploinsufficient for DDR-associated replication-born DNA damage in proliferating cells. We observed that acute *Nbn* monoallelic KO induces accumulation of DNA damage and apoptosis, likely due to an excessive accumulation of unrepaired DNA damage, in SMO agonist (SAG)-dependent cerebellar neurospheres (S-cNS) [156]. Notably, we also observed the activation of p53 but not alterations in cell proliferation or SHH pathway levels, as expected since p53 is a negative regulator of the SHH pathway [157,158,159], suggesting that *Nbn* haploinsufficiency could impair the DDR, at least initially, limiting p53 functions. Coherently, it was found that organoids from NBS patients undergo a delayed p53-mediated DNA damage response after bleomycin treatment [160], p53 is deregulated in NBS fibroblasts and NBS-iPSCs [131] and NBS-NPCs [161] compared with WT counterpart. Thus, damaged cells escaping from the p53-dependent death may become tumor-initiating cells. Coherently, *NBS1* mutations significantly associate with *p53* mutations in sporadic MBs [155]. Moreover, depletion of other DDR genes (such as *Lig4* and *Xrcc2* [162], *Xrcc4* [163], *Ku80* [164], *Brca2* [165] and *Parp1* [166]) in presence of *p53*KO induces SHH-MBs in mouse models supporting the idea that, in the presence of high levels of DNA damage, disabling p53 functions is a necessary condition to induce SHH-dependent cerebellar tumorigenesis. Notably, association between *NBS1* and *p53* mutations or pathway alteration (including *MDM2* amplification, *p14ARF* homozygous deletion and promoter methylation) were also found in other tumors [167,168,169], suggesting that simultaneous disruptions of NBS1 and p53 functions may constitute a novel genetic pathway in the pathogenesis of a subset of tumors, including MB. Coherently, a recent paper by Reuss et al. found that simultaneous *Nbn* and *p53* inactivation in neural progenitors triggers High-grade Gliomas [170].

Importantly, *Nbn* haploinsufficiency in mice is sufficient to induce a wide array of tumors affecting the liver, mammary gland, prostate, lung, and lymphocytes, but not the cerebellum [44], indicating that MRN complex haploinsufficiency needs “other factors” to transform this tissue. Thus, in our mouse model, SHH hyperactivation by SMOA1 expression seems to represent “the other factor”. Curiously, the SHH pathway carries out an inhibitory function on p53, favoring its degradation by MDM2, and SMO partially impairs p53-dependent apoptosis and cell growth inhibition in oncogene-expressing cells [171]. Therefore, from a different perspective, the SHH pathway may have contributed to *Nbn* haploinsufficiency-dependent tumorigenesis by inhibiting the p53-mediated anti-cancer barrier. In agreement with the idea that p53 has a particular role in the development of SHH-MB, the last is the MB subtype in which *p53* mutations are most enriched (21% of tumor samples), with more than half of the patients with SHH/p53 tumors having a germline mutation [172].

On the contrary, we demonstrated that complete *Nbn* loss fully prevents SHH-dependent MB development. These results are coherent with our previous data indicating that the MRN complex is necessary to counteract the RS induced by MYCN in proliferating GCPs [75,173], the cerebellar population from which MB arises. Of interest, the *SmoA1*/*Nbn*KO phenotype is essentially identical to the *Nbn*KO (not expressing SmoA1), suggesting that *Nbn* deletion is epistatic on the constitutive SHH pathway. Surprisingly, full deletion of *Nbn* correlates with low levels of SHH pathway expression/activity, in vivo and in vitro. Thus, since the dual role of the SHH pathway in both cerebellar development and transformation, we hypothesize that this regulation of the SHH pathway could contribute to the inhibitory effect on cerebellum development such as on MB tumorigenesis, observed after *Nbn* depletion. We speculate that the *Nbn*KO-induced impairment of the SHH pathway could depend on p53, functionally activated in cerebellar cells after *Nbn* full deletion [174] and widely recognized as an inhibitor of both Gli1 expression and activity [157,158,159]. According to a pivotal role of p53 in the *Nbn* depletion-dependent phenotype, p53 KO rescues cerebellar defects in *Nbn*KO mice [174].

Finally, we found that MRN proteins were increased in murine MBs samples compared to healthy cerebella, and similar results were found at the transcriptional level in human SHH-MB specimens compared to normal cerebellar tissues. Notably, we found comparable levels of NBS1 protein between *Nbn* heterozygous and WT full-blown MBs, suggesting the existence of a compensatory mechanism that potentiates the expression of the remaining allele.

Moreover, we demonstrated that *Nbn* depletion or pharmacological inhibition of the MRN complex by mirin induces DNA damage and apoptosis, impairing the growth of primary MB cells. These data are coherent with the idea that increased levels of MRN complex are beneficial for cancer cells experiencing high levels of RS, as those proliferating under the control of the SHH-MYCN pathway. Therefore, in this sense, our data support that the MRN complex plays an essential/oncogenic role, and that for this reason, it should be reasonably considered as a therapeutic target, potentially effective for the treatment of all tumors that present high levels of RS.

Therefore, by using a gene-dosage model of NBS1, we demonstrated that the role carried out by the MRN complex in cancer is strictly dependent on the amount of NBS1 protein expressed, which presumably determines how much RS-induced DNA damage is accumulated/induced, how much p53 can be functionally activated and, consequently, how many lesions can be tolerated without provoking apoptosis (Figure 5).

## 5. Future Perspective/Conclusions

Collectively our data and those found in the literature converge to the concept that MRN low levels or truncating or missense mutations in *MRN* genes may play an oncopromoting role as much as the overexpression of functional MRN proteins. Notably, they also reveal that a single wild-type allele (in hemizygous condition) may be sufficient to produce high levels of protein in full-blown tumors [120]. Moreover, in principle, a mutated allele may also be overexpressed. This aspect is not trivial because the totality of studies correlating the MRN complex with prognosis achieves this conclusion by using MRN protein or transcriptional analysis that, obviously, are not able to distinguish between wild type and MRN defective tumors. We believe that this aspect may contribute to the conflicting results emerging from the clinical studies on the role of the MRN complex as prognostic and predictive factor. Indeed, although the frequency of pathogenic variants within the MRN complex is about 1% or less in all cancer patients, it results relatively high in some specific tumor backgrounds (see above). Moreover, the number of carriers may be underestimated due to the fact that we do not know the clinical significance, for hundreds of variants within *MRE11A* (https://www.uniprot.org/uniprotkb/P49959/entry; accessed on 1 June 2023), *RAD50* (https://www.uniprot.org/uniprotkb/H7C0V2/entry; accessed on 1 June 2023), and *NBS1* (https://www.uniprot.org/uniprotkb/O60934/entry; accessed on 1 June 2023) genes. Therefore, the genetic/functionality state of the MRN complex should be considered when we assign prognostic value to the MRN complex expression. For all these reasons, we speculate that a combined analysis including MRN gene and protein levels characterization would be more useful to understand the biological features of tumors and even drive towards the best therapeutic choice.

*MRN* gene alterations that even partially impair MRN complex functionality may contribute to cancerogenesis by altering ATR/ATM-dependent anti-cancer barrier and creating an environment that selects for mutations or epigenetic silencing of checkpoint genes (e.g., *p53*) and promotes genome instability. Thus, *MRN* status can correlate with responses to specific DNA-damaging therapy. Indeed, *MRN* defective tumors, such as HR and NHEJ deficient tumors, may be successfully treated with agents that produce DSBs using a synthetic lethality-based approach, such as with the PARP inhibitors. These last should be preferred to the standard therapy also because patients with *MRN* germinal defects may suffer from chemo-radio-therapy-induced toxicity [154] and target therapy reduces the need for highly toxic combination chemotherapeutic regimens. Moreover, *p53* deficiency or mutation enhances the cytotoxicity of PARP inhibitors in various tumors [175] and mutations in the *NBS1* gene often correlate with *p53* mutations [167,168,169], further supporting the use of PARP inhibitors in these backgrounds.

On the contrary, the functions of the MRN complex in DDR and RSR make its expression indispensable for cancer cell survival. Indeed, MRN complex overexpression correlates with enhanced DDR and drug resistance in most cancers. Importantly, since several chemotherapy drugs induce RS [176,177], chemotherapy-treated and chemo-resistant cells could be considered automatically dependent on the MRN complex. In all these contexts, increasing evidence from in vitro and preclinical studies indicates that MRN complex targeting can be effective. However, currently, we cannot evaluate this approach in patients because drug development investigations targeting the MRN complex genes are limited. Indeed, despite some drugs that target the endonuclease (PFM01 and PFM03 [13]), or exonuclease activity (mirin [84] and PFM39 [13,178]) of MRE11 were identified, they do not have pharmacological properties permitting their use in clinical.

Importantly, it should be recalled that all components of the MRN complex are essential for the viability of mammalian cells [2,3,4] and it is believed that this may depend on the role of the MRN complex in preventing and repairing replication-intermediates, avoiding cell death [179]. Consequently, also normal tissues may be sensitive to MRN complex targeting, especially whether it is obtained by using inhibitors of the MRE11-dependent exonuclease activity [180]. Therefore, future studies should be focused on finding pharmacological inhibitors of the MRN complex coupled with a delivery system that transports them specifically to cancer cells.

Despite we cannot target the MRN complex currently, we could consider its overexpression in cancer as a biomarker of elevated RS that may predict sensitivity to inhibiting kinases that coordinate the DNA damage response with cell cycle control, including ATR, CHK1, WEE1 checkpoint kinases [181]. Indeed, for these molecules, pharmacological inhibitors are already available and currently under consideration in several clinical trials [182].

In conclusion, we believe that at least some of the roles that the MRN complex plays in cancer are less than controversial and may represent useful instruments to improve the prognosis of many cancer patients. Indeed, given the roles of MRN complex in DDR/RSR, *MRN* genes haploinsufficiency might be used as a marker that predicts a good response to a therapy based on the synthetic lethality, as well as the overexpression of functional MRN proteins might be exploited to select tumors that may respond to the targeting of the RSR (Figure 6). However, currently, we are unable to convert our knowledge in this field into clinical practice due to the lack of an operative model that, combining genetic and biochemical analysis, permits to distinguish between MRN defective and proficient tumors. A possible start point to achieve this goal is by the realization of a standard practice including (Figure 6): (i) the full sequencing of the three *MRN* genes in tumor tissues, coupled with informatics tools predicting the level of MRN complex dysfunction in silico; (ii) find reliable biomarkers of MRN complex activity in DDR and RSR to validate in silico forecast; (iii) the evaluation of protein levels and localization (nuclear versus cytoplasmic) of all the three MRN proteins using a standardized protocol of immunohistochemistry (IHC) provided with an automated intensity-scoring system, calibrated on normal tissue. Future efforts should be intensified in this sense to validate the prognostic/predictive significance of the MRN complex in cancer.

## Figures and Tables

**Figure 1 cancers-15-03599-f001:**
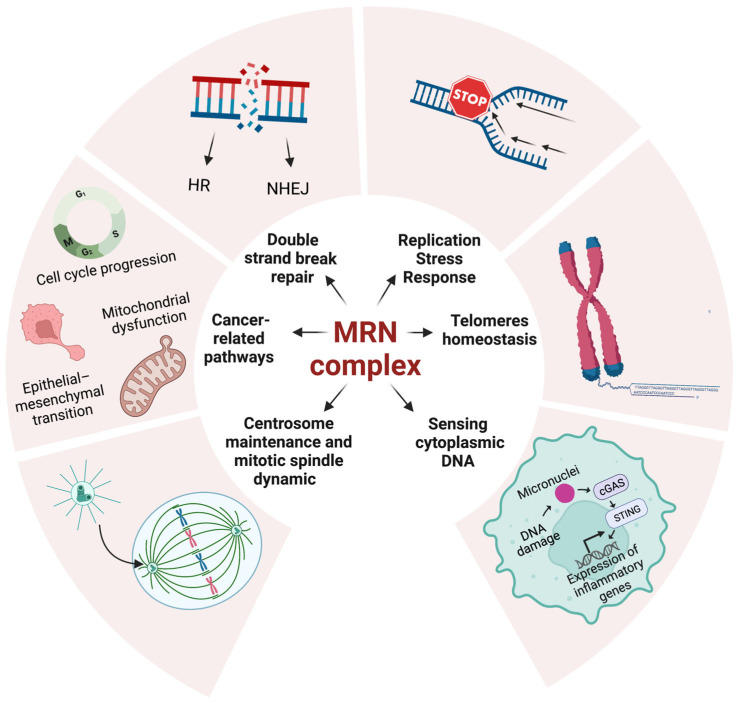
The multiple roles of the MRN complex in cell homeostasis. The functions that the MRN complex plays in healthy cells may be altered in MRN-defective tumors and hyperstimulated in MRN-proficient cells. The figure was created with Biorender.com.

**Figure 2 cancers-15-03599-f002:**
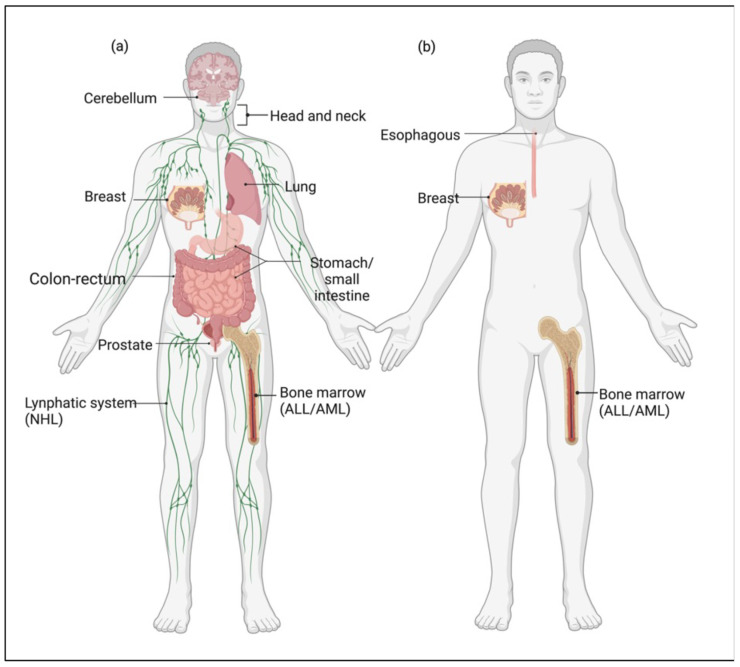
*MRN* gene mutations in cancer. Organs and tissues where *NBS1* (**a**) and *RAD50* (**b**) heterozygous germline mutations increase cancer risk. The figure was created with Biorender.com.

**Figure 3 cancers-15-03599-f003:**
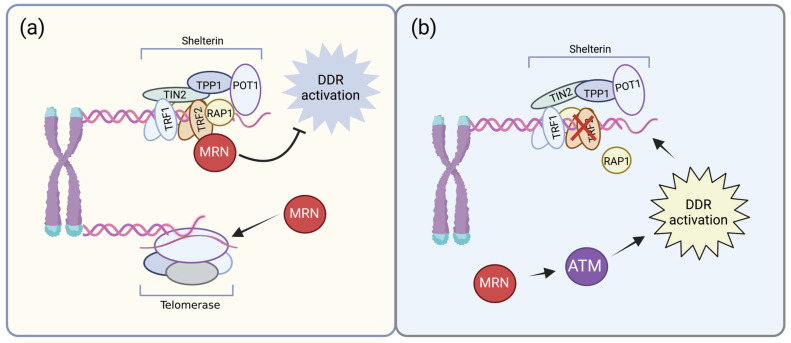
MRN complex in telomere homeostasis. (**a**) MRN specifically associates with the TRF2 component of the Shelterin complex and participates in the formation of t-loops, contributing in to protect chromosome ends from DDR factors. Moreover, MRN facilitates telomerase activity at telomeres. Whether MRN make it by modifying telomere ends, opening up the t-loop, altering chromatin structure, or by directly associating with telomerase remains to be determined. (**b**) On the other side, at dysfunctional telomeres (exemplified in the figures by TRF2 depleted telomeres), the MRN complex is essential to mediate DDR activation to uncapped telomeres. The figure was created with Biorender.com.

**Figure 4 cancers-15-03599-f004:**
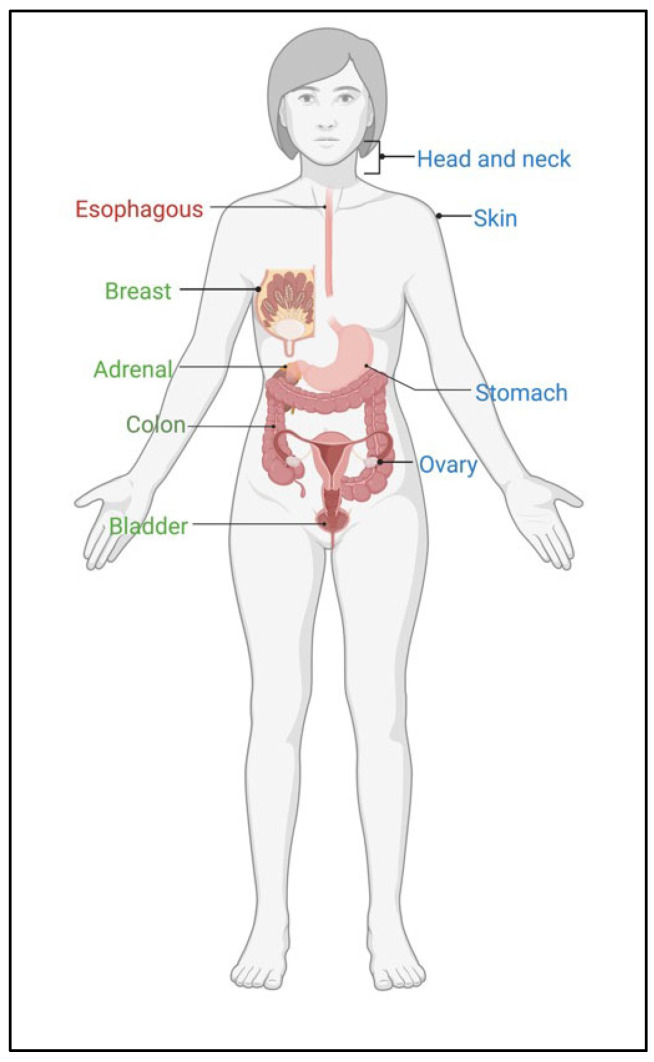
MRN complex expression and cancer prognosis. In red and in blue, the tumor backgrounds in which high levels of the MRN complex correlate with good and poor prognosis, respectively. In green, the tumor backgrounds where, in dependence on the publication, high MRN complex expression seems to correlate with poor or good prognoses. The figure was created with Biorender.com.

**Figure 5 cancers-15-03599-f005:**
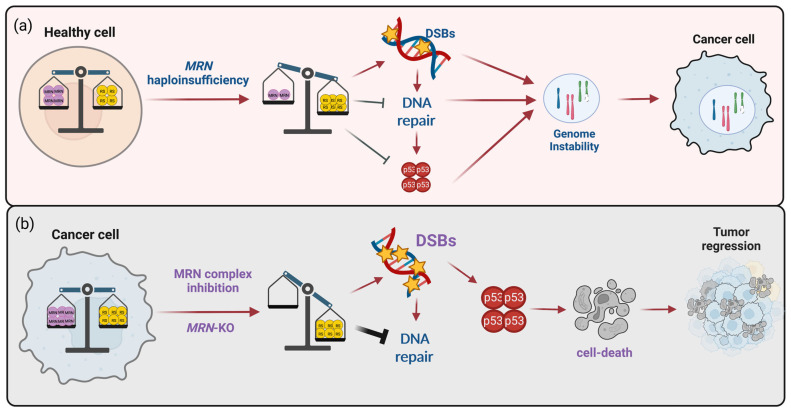
How MRN expression/activity may affect tumor initiation and progression. (**a**) In healthy cells, MRN complex mitigates physiological RS. In the presence of MRN haploinsufficiency, RS increases favoring DSB accumulation. Contextually, the DNA repair pathway and p53 functions are defected, contributing to a further increase DSBs in a context permissive to cell survival. This condition may lead to genome instability favoring the selection of tumor-initiating cells. (**b**) In full brown cancer cells, the MRN complex counteracts the oncogene/therapy-induced RS. Here, MRN complex targeting increases RS-induced DSBs and impairs their repair. Thus, the high levels of DSBs become incompatible with cell survival, favoring tumor regression. The figure was created with Biorender.com.

**Figure 6 cancers-15-03599-f006:**
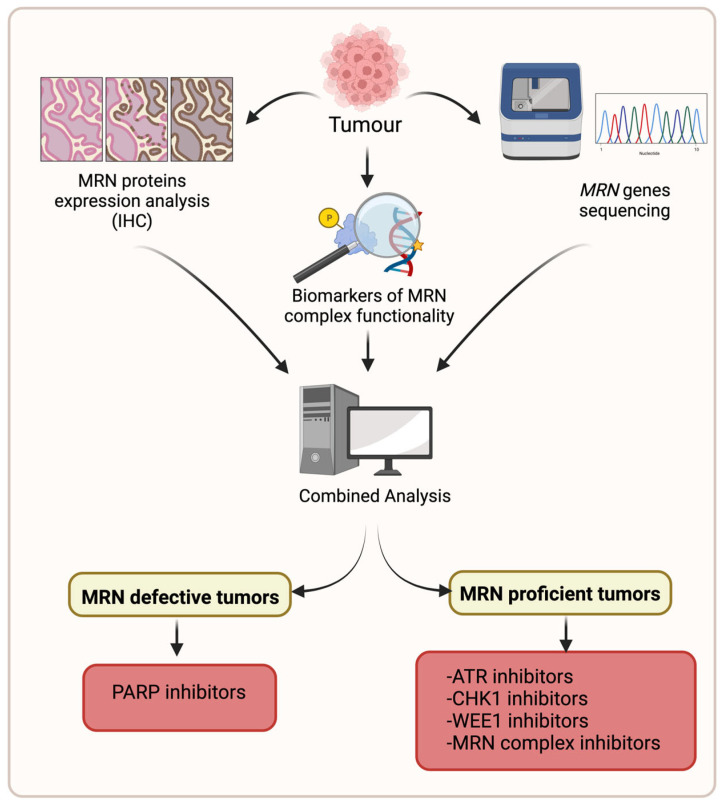
How to use MRN complex expression/functionality as predictive marker in cancer. We believe that a big part of the conflicting studies on the role of the MRN complex in prognosis depends on the fact that they do not distinguish between MRN defective and proficient tumors. To overcome this limit, we propose to introduce a new medical practice in which genetic, biochemical, and in silico analyses of the MRN complex are combined to predict the best therapeutic choice in prospective studies. It is expected that MRN-defective tumors may result in sensitivity to PARP inhibitors, while MRN overexpressing tumors may be effectively treated with drugs targeting the RSR. The figure was created with Biorender.com.

**Table 1 cancers-15-03599-t001:** Frequency of *NBS1* heterozygous germline mutations among tumor patients and control subjects. n.s.: not significant.

Cancer	Mutation	Number of Patients with Cancer	% of Cancer Patients with *NBN* Mutation	Number of Healthy Subjects	% of Healthy Subjects with *NBN* Mutation	Statistical Analysis	PubMed IDentifier
Acute Lymphoblastic Leukaemia	p.I171V	46	10.7	2400	0.5	0.0004	24093751
Acute myeloid leukemia	p.I171V	32	6.3	2400	0.5	0.0001	24093751
Breast (familial)	c.657_661del5	80	1.3	530	0.6	n.s	12845677
Breast	c.657_661del5	173	1.2	344	0	0.046	15578693
	c.657_661del5	224	1.8	1620	0.6	n.s.	15185344
	c.657_661del5	150	3.7	530	0.6	0.037	12845677
	c.657_661del5	562	2.0	1620	0.6	0.0107	16770759
	c.657_661del5	700	0.7	344	0	n.s.	15578693
Colorectal	c.657_661del5	234	1.3	1620	0.6	n.s.	15185344
	I171V	131	2.3	600	0.2	0.0196	18280732
	IVS11+2insT	472	0.6	2348	0.08	0.02	18056440
	R215W	234	1.3	1620	0.2	0.0472	15185344
Gastric cancer	IVS11+2insT	472	0.4	2348	0.08	0.0001	18056440
Gastrointestinal lymphoma	c.657_661del5	37	10.8	1620	0.6	0.0002	16998789
Head and neck	I171V	81	6.2	600	0.2	0.0001	18280732
Larynx cancer	I171V	176	2.3	500	0.2	0.0175	17894553
Lung	c.551A>G	453	3.8	2400	0.5	<0.0001	26722329
	c.657_661del5	453	0.7	2090	0.2	n.s.	26722329
	IVS11+2insT	532	0.4	2348	0.08	n.s.	18056440
Medulloblastoma	c.511A>G	104	3.8	4227	1.3	0.0241	19908051
	c.657_661del5	104	2.9	12,484	0.6	0.0028	19908051
Melanoma	c.657_661del5	376	0.3	866	0.1	n.s.	17496786
	c.657_661del5	80	2.5	530	0.6	n.s.	12883362
	c.657_661del5	105	3.8	1620	0.6	0.0081	15185344
Non-Hodgkin Lymphoma	c.657_661del5	109	0	984	0.3	n.s.	10848790
	c.657_661del5	42	4.8	1620	0.6	0.0351	15185344
	c.657_661del5	228	3.5	1620	0.6	0.0001	16998789
Prostate (familial)	c.657_661del5	56	9.0	1500	0.60	<0.0001	14973119
Prostate	c.657_661del5	305	2.2	1500	0.60	0.01	14973119

**Table 2 cancers-15-03599-t002:** Frequency of *RAD50* heterozygous germline mutations among tumor patients and control subjects. n.s.: not significant; n.d.: not determined.

Cancer	Mutation	Number of Patients with Cancer	% of Cancer Patients with *RAD50* Mutation	Number of Healthy Subjects	% of Healthy Subjects with *RAD50* Mutation	Statistical Analysis	PubMed IDentifier
Acute Lymphoblastic Leukaemia + Acute myeloid leukemia	-	220	-	504	0.006	0.0019	24093751
Breast	687delT	317	2.52	1000	0.6	0.008	16474176
	-	7657	0.07	5000	0.02	n.s.	29726012
	687delT	590	0.5	560	0.2	n.s.	16385572
Breast/Ovarian	687delT	151	1.3	1000	0.6	n.d.	14684699
Esophageal squamous cell carcinoma	p.K722fs	2088	0.14	2342	0	0.032	34572942
	p.Q672X/p.K722fs	2088	0.19	2342	0	0.01	34572942
FH^+^ Esophageal squamous cell carcinoma	-	372	1.1	19,954	0.15	0.0033	34572942

**Table 3 cancers-15-03599-t003:** Frequency of *MRE11* heterozygous germline mutations among tumor patients and control subjects. MSI: microsatellite instable; * indicates a truncating mutation; n.s.: not significant; n.d.: not determined.

Cancer	Mutation	Number of Patients with Cancer	% of Cancer Patients with MRE11 Mutation	Number of Healthy Subjects	% of Healthy Subjects with MRE11 Mutation	Statistical Analysis	PubMed IDentifier
Breast	-	75,818	0.05	111,326	0.05	n.s.	31406321
	p.E506 *	1925	0	2287	0.002	n.s.	33510186
	p.R364X	1	n.d.	100	0	n.d.	28559769
Colorectal	-	1006	0.3	1609	0	n.d.	27329137
	-	49	0.84	n.d.	n.d.	n.d.	15048091
Endometrium	-	14	7	n.d.	n.d.	n.d.	15048091
	p.E506 *	367	0	2287	0.002	n.s.	33510186
Gastric MSI	poly(T)11	27	0.81	n.d.	n.d.	n.d.	15319296
Hereditary Breast and Ovarian Cancer	p.R305W	151	0.3	1000	0	n.d.	14684699
	-	708	0.4	n.d.	n.d.	n.d.	24549055
Medulloblastoma MSI	poly(T)11	4	0.25	n.d.	n.d.	n.d.	19179424
Ovary	p.E506 *	341	0	2287	0.002	n.s.	33510186
	-	5	0.2	n.d.	n.d.	n.d.	15048091
Stomach	-	1	1.0	n.d.	n.d.	n.d.	15048091
Ureter	-	1	0	n.d.	n.d.	n.d.	15048091

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
