# Peer review of "The Multiple Faces of the MRN Complex: Roles in Medulloblastoma and Beyond"

_cancers, 2023, doi:10.3390/cancers15143599_

Round 1

Reviewer 1 Report

This is a well-written manuscript titled “The Multiple Faces of the MRN Complex: Roles in Medulloblastoma and Beyond” that discuss the possibility to use genomic and molecular characterization of the MRN complex in clinical practice to improve therapeutic strategies. The figures/ tables/ schemes are appropriate and align with the text and in the present review, the author has comprehensively summarized how alterations in the MRN complex impact the pathophysiology of cancer and discusses their recent discoveries on the gene-dosage-dependent effect of NBS1 in Medulloblastoma. This manuscript also identified a gap in knowledge and will be of good interest to the scientific community. The manuscript can be accepted in its present form with minor revisions.

1.      The figures/ tables/schemes are self-explanatory and nicely illustrated in the present review. It will be helpful for the reader if the author adds some figures explaining the mechanism of action of the MRN Complex in Telomere Homeostasis.    

Dear Editor,

The manuscript is well-written and easy to understand. It needs minor proofreading for a spelling check. 

Regards,

Ajit 

Reviewer 2 Report

  • A brief summary: A comprehensive literature by Pertroni et al. sheds light on the impact of the MRE11/RAD50/NBS1 (MRN) genes and the overexpression of MRN proteins on cancer initiation and progression. In this review, the authors meticulously examine relevant studies in this area, drawing upon their recent publication that investigates the dual role of NBS1 as both a haplo-insufficient tumor suppressor and an essential gene in SHH-medulloblastoma. Furthermore, the authors explore the potential application of genomic and molecular characterization of the MRN complex in clinical practice to enhance therapeutic strategies. They also emphasize future research directions and the challenges associated with this field.
  • General concept comments

ü  Review: The review provides a thorough examination of the diverse roles played by the MRN complex in the development of cancer, presenting a comprehensive understanding of this topic. Within the review, the authors investigate how alterations in the MRN complex influence the pathophysiology of cancer, taking into account their recent discoveries regarding the gene-dosage-dependent effects of NBS1 in medulloblastoma. These new insights aim to assess the potential use of the MRN complex as a prognostic or predictive marker, as well as a therapeutic target, for the treatment of cancer patients in the future.

ü  Specific comments: The review is well written in fluent English, highly informative and suggest its acceptance in Cancers.

Reviewer 3 Report

1. Overall the review was very nice to read. I have minor comments. First the term hypomorphic is rather use to describe the mutations in the human diseases associated with the mutations in MRE11A, NBS1 and RAD50. For the sporadic tumours i will rather used the word truncated, in frame or out of frame that define better these types of mutations and are not coinfusing.

2. i know it is always complicated with the MRN complex and the gene nomenclature but it should be throughout the text MRE11A for the gene/MRE11 for the protein, NBN for the gene (which i agree the nomenclature is not good) therefore better to use Nbs1 but explaining the offical nomenclature.

3. there are some typos that have to be checked indeed it is for example XRCC1-deficient rather than XRCC1 deficient.

4. maybe i missed them but there are several references missing: on MRE11 and NBS1 on Top1cc and top2cc removal. On the top, the last paper of ISG15 and Nbs1 is missing. Finally, it would be interesting to include the recent Reuss et al, 2023 which provides additional evidences to the claims of the authors.

There are some sentences that need to be checked for example page 5 "MRE11 heterozygous germline mutations seem not predispose to cancer"

or in figure 2. MRN genes mutations in cancer. Just double checked the text there are minors typos text that i can not list all here.
